# Forest Therapy Alone or with a Guide: Is There a Difference between Self-Guided Forest Therapy and Guided Forest Therapy Programs?

**DOI:** 10.3390/ijerph18136957

**Published:** 2021-06-29

**Authors:** Jin-Gun Kim, Won-Sop Shin

**Affiliations:** 1Korea Forest Therapy Forum Incorporated Association, Cheongju 28644, Korea; K64804171@gmail.com; 2Department of Forest Sciences, Chungbuk National University, Cheongju 28644, Korea

**Keywords:** self-guided forest therapy, guided forest therapy program, healing factor, health benefit

## Abstract

There are generally two types of forest therapy. One is to walk or view the forest alone without a guide, and the other is to be accompanied by a guide. This study aimed to investigate the healing factors and health benefits of self-guided forest therapy and guided forest therapy programs and examine the differences in characteristics between interventions. Thirty-seven undergraduate students participated in a randomized experiment (19 in the self-guided forest therapy and 18 in the guided forest therapy program). Data were collected from 111 self-reported essays after each intervention (three essays per person). Results revealed that the forest healing factors contained four categories in common: auditory element, visual element, tactile element, and olfaction element. Forest therapy’s health benefits included five categories in common: change of mind and body, introspection, change of emotion, cognitive change, and social interaction. Among the typical differences, the self-guided forest therapy group mentioned more keywords related to introspection than the guided forest therapy program group. On the other hand, the guided forest therapy program group mentioned more keywords associated with the change of emotion and social interaction than the self-guided forest therapy. Our findings show that self-guided forest therapy provides an opportunity for self-reflection to focus on and think about one’s inner self. On the other hand, guided forest therapy programs provide positive emotional changes and promoting social bonds through interaction with others. Therefore, because the effects that can be obtained vary depending on the type of forest therapy, participants can utilize forest healing to suit the desired outcomes.

## 1. Introduction

The world has become an urban society, with over half the world’s populations living in urbanized areas. Urbanization, defined as the relative increase of the urban population as a proportion of the total population [1], is a demographic movement and includes social, economic, and psychological changes [2]. Historically, urbanization was related to positive economic development and provided many opportunities such as education and good infrastructure [3], but overexposure to stress caused mental health problems [4,5]. Urbanization negatively affects mental health by influencing increased stressors and factors such as overcrowded and polluted environments, high noise levels, higher crime rates, reduced social support, and lack of calming scenery [6]. Therefore, it is crucial to find ways of promoting mental health recovery in daily life.

Connecting with forests has been increasingly recognized as an effective method of dealing with mental problems [7,8]. The term “forest bathing,” derived from Japan, is defined as being in a forest environment to restore balance psychologically and physiologically and absorb its atmosphere [9]. In later years, the term “forest bathing” developed into “forest therapy,” which is the medically proven health effects of exposure to forest environment [10]. The Korea Forest Service (KFS) has legitimized the concept of forest therapy and defined forest therapy as “immune-strengthening of the human body and health-promoting activities by utilizing various elements of nature, such as scent and landscape [11].” Based on the empirical research evidence, the Korea Forest Service (KFS) facilitated forest therapy to utilize forests for enhancing people’s health and quality of life.

Many studies demonstrated forest environments promote humans’ mental and physical health in many ways. For example, spending time in a forest environment reduces pulse rate and blood pressure [12,13], sympathetic nervous activity [14,15], and levels of salivary cortisol [16,17], and increases parasympathetic nervous activity [18,19] and NK cell activity, which relate to the immune system [20,21]. In addition, many studies have shown the beneficial effects on mental health of being in the forest. Those studies indicated that being in the forest can reduce negative emotion [22,23], depression [24], anxiety [25], and stress [26], and increase positive emotion [22,23,24] and self-esteem [27].

There are generally two types of forest therapy. One is to walk or view the forest alone without a guide and the other is to be accompanied by a guide. Namely, when people visit the forest to improve their health, people participate in forest walking or viewing forests individually or participating in a forest therapy program, guided by the therapist with a license. A forest therapy program is a set of structured activities and cognitive-behavioral therapy-based interventions using various elements of the forest environment to mitigate stress and to promote health [28]. Many previous studies have reported the effects of the two types of activities in forest therapy.

For individual visits to forests without a guide, numerous studies have demonstrated the effects of walking or viewing the forest alone in relieving stress levels and inducing psychological relaxation [14,22,29,30]. For example, Park et al. [22] showed that walking and viewing forests improved emotional state, leading to psychological relaxation. Song et al. [30] also reported that participants who walk in a forest increase parasympathetic nervous activity and decrease sympathetic nervous activity and heart rate compared to walking in a city environment. In addition, those who walked in a forest improved positive emotions such as vigor and reduced the anxiety dimension of the STAI and negative emotions such as tension/anxiety, depression/dejection, anger/hostility, fatigue, and confusion. Research has also shown that exposure to nature, such as forests, improves cognitive tasks requiring direct attention. For example, Tennesen and Cimprich [31] reported that students who could see the natural environment through dormitory windows performed better on tasks requiring concentration than those who did not. A similar study by Taylor et al. [32] showed that children who see urban forests around apartments perform better in memory, impulse control, selective attention, and concentration. Berto [33] also reported that participants who looked at pictures of the natural environment had more sustained attention than those who looked at images of the urban environment. Similarly, walking in a natural environment has shown advantages in language work memory and cognitive control relative to walking in an urban environment [34,35,36].

Previous studies also showed that being in the forest without a guide can relax the body and mind and provides an opportunity to look back on oneself. For example, Sonntag et al. [37] reported people diagnosed with exhaustion disorder who spent two hours alone in the forest increased happiness and were more relaxed, harmonious, alert, and clear-headed compared to before. Furthermore, those who participated also reported that they began to take charge of their lives after a while, and that they began to plan to change their lives after a very long time. Korperla [38] and Hammitt [39] also showed that solitude in natural places could regulate negative moods. Being in a restorative natural environment that deviates from everyday life can provide an opportunity to look at life in a different context, connecting concepts that could not have been created before. The privacy available in the forest offers a unique setting to utilize self-reflection. Self-reflection has been described as involving “active, persistent, and careful consideration” [40]. Boud et al. [41] describe it as “those activities individuals engage in to explore experiences.” With the many distractions experienced in modern life, reflection, self-awareness, and integration are very important. The self-reflection chance offered by nature might be maximized when spending time in the forest autonomically rather than controlled by guides. It could mean that being self-guided in a forest setting might provide an opportunity to think about the current situation and allow more positive thinking. During this process, the environmental elements of forests, which can be represented by the scenery, smell, and sound of the forest, can stimulate the five senses of humans to sympathize with natural environments in the forest.

Many previous studies have reported that the guided forest therapy programs can have a substantial physiological and psychological response. For example, forest therapy program provided a decrease in pulse rate [42], blood pressure [42,43], and cortisol [44], reduced negative emotion, and improved positive emotion [45,46]. Furthermore, many studies have investigated elderly individuals and adults at risk of stress and lifestyle-related diseases such as high blood pressure, diabetes, and depression. For example, Ochiai et al. [47] conducted a forest therapy program for middle-aged males with high normal blood pressure. The results showed that participants’ systolic and diastolic blood pressure, urinary adrenaline, and serum cortisol were significantly reduced after forest therapy. Participant’s negative emotion scores such as tension/anxiety, confusion, and anger/hostility, and total mood disturbance scores were reduced. Song et al. [48] reported that systolic and diastolic blood pressure decreased during the forest therapy program, and these decreases were maintained for five days. Shin et al. [49] also reported that the two-day forest therapy program provided significantly positive changes in workers’ job stress and moods. In addition, the effects of a forest therapy program for people who have mental health problems have been investigated. Bielinis et al. [50] conducted a forest therapy program for people with post-traumatic stress disorder or experiencing stress. The results showed that in people with psychotic disorders, the forest therapy program increases the vigor trait. In contrast, in patients with affective disorders, the forest therapy program was more effective in improving confusion and depression/dejection traits. Woo et al. [51] also found that patients with a major depressive disorder showed more significant improvement in depression and quality of life after the forest therapy program. Hong et al. [52] reported that psychiatric outpatients who participated in a three-day forest therapy program reduced depression, anxiety, and anger, and improved quality of life compared to before.

The forest therapy program is also helpful in social health. Especially for children, previous studies showed that forest therapy programs could improve sociality [53,54], social competence [55,56], and social development [57,58], and reduce problem behavior [59]. For example, Jang et al. [58] reported that an after-school forest therapy program for infant participants effectively improved pro-social behavior and expression efficiency. In addition, Kim and Kang [57] showed that cooperative play in the forest improves young children’s social skills and happiness. Choi et al. [55] also reported that children who played in the forest enhanced their sense of social competence and happiness relative to playing in the classroom. Lee et al. [60] reported that middle-aged women who participated in the urban forest therapy program acquired knowledge about nature and forming emotional bonds with one another. According to Rogers [61], the process of psychological change formed by building social relationships in forest environments was similar to developing rapport. Moreover, the participants began to share their negative feelings after positive feelings developed through mutual understanding and regard. Overall, findings seem to suggest that a guided forest therapy program brings about positive impacts on individuals’ psychological and physiological functioning as well as social skill. Guided forest therapy programs are led by trained forest therapy guides and involve various physical activities that would not be possible to engage in alone, although slight variation may exist between programs. Guided forest therapy programs may share a variety of emotions and help social interactions by experiencing various activities with participants under the guidance of therapists.

However, few studies have discussed and investigated the difference between being in the forest alone and guided forest therapy programs. Staats and Hartig [61] investigated social context influences (alone or in the company of a friend) and psychological restoration on preferences for natural and urban environments. It was found that the company of a friend created a feeling of safety that could enable restoration. However, when security is ensured, they found walking alone enhanced more restoration than being in company. Igawahara et al. [62] investigated participants’ physiological and psychological effects in the forest therapy and walk-alone program. The results showed that subjects who participated in the forest therapy program found it more therapeutic and relaxing than those who participated in the walk-alone program. Lim et al. [63] reported that comparing the guided and unguided conditions, there were no significant differences in the change in nature connectedness, mood, or heart rate.

Even though a few studies have examined the differences between guided forest therapy program and self-guided forest therapy, it remains unclear whether and how guided or unguided forest therapy could contribute to participants’ health. There is still a general lack of research on this area. Therefore, this study aimed to investigate the healing factors and health benefits of self-guided forest therapy and guided forest therapy programs and examine the differences in characteristics between interventions.

The following research hypotheses were formulated in the study:The self-guided forest therapy group will mention more varied forest healing factors than those who participated in the guided forest therapy program.The self-guided forest therapy group will mention reflecting on oneself more than those who participated in the guided forest therapy program.The guided forest therapy program group will mention more varied emotional changes than those who participated in the self-guided forest therapy.The guided forest therapy program group will mention more social interaction than those who participated in the self-guided forest therapy.

## 2. Materials and Methods

### 2.1. Participants

Thirty-seven undergraduate students of Chungbuk National University (23 males, 14 females) participated in this study, and their mean age was 21.4 ± 1.3 years. Participants were recruited by posting notices throughout the university building to recruit volunteers. No incentives were provided to the volunteers. The inclusion criteria for recruiting the participants who were eligible for the study were: (1) no diagnosis of reaction to severe stress and/or a depressive episode; and (2) could not be suffering from any drug or alcohol abuse. The participants were randomly assigned into two groups (i.e., 19 in the self-guided forest therapy group and 18 in the guided forest therapy program group). The experiment was conducted during the 2nd semester of 2019 (September–November). A total of eight sessions of self-guided forest therapy and guided forest therapy program were performed. Before starting the experiments, we explained the aims and procedures of the study to the participants and obtained a voluntarily signed agreement. This study was approved by the Institutional Review Board of Chungbuk National University (IRB number: CBNU-201910-SB-945-01).

### 2.2. Experimental Sites

The field experiment site was conducted in the Chungbuk National University campus forest in Korea. The size of the campus forest is about 315,000 square meters. The campus forest mainly contains metasequoia (*Metasequoia glyptostroboides*), cypress (*Chamaecyparis pisifera*), and mixed forest species (pitch pine, oak, chestnut), and their ages range from 40 to 90 years. Therefore, the study area was suitable for conducting forest therapy activities regarding accessibility, distribution of various vegetation, and gentle slope. During the eight sessions in the experiment, the weather was pleasant and not raining, with a mean temperature of 16.2 °C ± 1.3 °C.

### 2.3. Experimental Design

#### 2.3.1. Guided Forest Therapy Program

The guided forest therapy program was provided from September to November of 2019 and consisted of one hour and a half per week for eight weeks. A trained forest therapist delivered each session. The purpose of the guided forest therapy program was to reduce stress and improve self-esteem and happiness for participants. During the eight sessions of the program, participants engaged in each of the eight sessions together. They performed various activities, such as familiar with the forest, clapping exercises, forest folk dance, forest orienteering (using natural objects to solve group mission), forest walking, photo healing (taking pictures of nature and story-telling), talking to nature, and a rope game (see Table 1). The forest therapy activities were developed in consultation with researchers in forest therapy/forest recreation and forest therapists.

#### 2.3.2. Self-Guided Forest Therapy

The participants were instructed to perform forest activities for eight sessions (one hour-long session per week). We designed the forest activities road to allow participants to perform forest activities voluntarily without guidance by therapists. The forest activities interventions were provided from September to November of 2019. During the eight sessions of the intervention, participants engaged in each of the eight sessions together. They were asked to walk the well-designated route in the campus forest, visit specific locations where explained panels on forest activities were installed, and return to the starting point at an appointed time. The length of the route was 1.7 km long and was a loop trail. The self-guided forest therapy activities were designed and distributed for the same purpose as the forest therapy program described earlier. The self-guided forest therapy activities were picked up based on consultation from experts in the forest therapy field, including researchers and practitioners. Accordingly, five forest activities were selected to apply as the self-guided forest therapy activities: stretching, respiration, walking, meditation, and exercise.

### 2.4. Data Analysis

#### 2.4.1. Data Collection

The analysis data of this study were self-reported essays written by the subjects after the eight-session guided forest therapy program and self-guided forest therapy activities. The topics of essays used in this study were related to health benefits or changes that they felt after forest therapy. The text format was free prose which was within one page of A4 sheets in length. Participants were asked to write the essay on the day each session ended. Each participant was asked to submit three essays during the eight sessions. Essays were collected three times after completing one to three sessions, four to six sessions, and seven to eight sessions. As a result, 111 essays were analyzed, with 57 essays submitted by 19 self-guided forest therapy groups and 54 essays submitted by 18 guided forest therapy groups.

#### 2.4.2. Qualitative Data Analysis

The collected data were analyzed using NVivo 12.0 to add reliability to the analysis of data through transcription, coding, and analysis, and analysis of the discovery contents of the topic presented by Kim [64]. The NVivo 12.0 qualitative data analysis software is a tool created for social science mixed-method research that is well suited for coding and analyzing qualitative data. For data analysis, experience essays were converted into text documents, and the text data stored in NVivo 12.0 was repeatedly read. Open coding was conducted to find an influential group of words in the data and assign code names, then reclassify them by subject. Subsequently, in-depth coding was performed to reduce the amount of coding and to clarify its meaning. The research paradigm was based on the post positivism [65] as it enabled us to focus on the practical application of our research using data collection and the analytic approach best suited to testing our research hypotheses. This approach was underpinned by ontological critical realism and epistemological constructionism [66]. The research method was based on content analysis, which as a research method is a systematic and objective means of describing and quantifying phenomena [67,68,69]. To increase the reliability of the study, a triangulation technique was conducted. In addition, two experts in forest recreation and forest therapy and analyzed the essays to expand the scope of the data. The analysis was carried out in four stages of open coding with forest therapy researchers. The two authors (J.G., W.S.) read a sample of transcripts, continuously compared data, codes, and categories during the coding process, and discussed the emerging themes until they reached agreement. The sequence of analysis was as follows. First, the overall feeling and perception of forest therapy activities were derived by repeatedly reading the data. Second, using line-by-line analysis, each sentence was read line-by-line. All sentences containing words or contents related to forest healing factors and health benefits were coded in each case. The written essays were initially sorted into codes based on participants’ language; code names were developed to closely reflect the particular wording used by participants and were based on the most frequently mentioned words/phrases. When possible, forest healing factors related codes were phrased as nouns (e.g., bird sounds rather than listen to bird sounds), highlighting the elements of the forest environments that might bring a person to the health benefits. In addition, health benefits-related codes were phrased, when possible, as descriptive statements to keep with the way participants used language to describe their experience during forest therapy activities. For example, health benefits related to positive feelings were coded as “positive feelings were improved” rather than “positive feelings”. Third, focusing on recurring themes of forest healing factors and health benefits, concepts were derived by similar grouping themes. Codes were clustered into themes. Each theme was found according to the forest healing factors and health benefits. Themes were derived from participants’ language (e.g., most frequently mentioned word/phrase within a cluster of codes) and findings from previous studies [70].

Therefore, overall, forest healing factors were restructured into four categories (auditory, tactile, visual, olfactory) and health benefits into five categories (change of mind and body, change of emotion, social interaction, cognitive change, and introspection).

#### 2.4.3. Quantitative Data Analysis

The analysis of this study was to quantify forest healing factors and health benefits derived through qualitative analysis of essays submitted after the interventions. Accordingly, quantitative data analysis was conducted for quantitative comparison between groups according to the frequency of words related to forest healing factors and health benefits.

The statistical analyses were performed using SPSS 18.0 Windows (SPSS, Chicago, IL, USA). Pearson’s Chi-squared test of independence was used to determine the difference between forest healing factors and health benefits derived after experiencing self-guided forest therapy and guided forest therapy programs. All statistical tests used a *p*-value < 0.05 as the significance level.

## 3. Results

### 3.1. Quantitative Comparison of Forest Healing Factors in Self-Guided Forest Therapy and Guided Forest Therapy Program

The types of forest healing factors derived from open coding in this study were shown in 4 categories and 17 keywords: auditory elements (birds, wind sounds, fallen leaves, bug, etc.), visual elements (natural colors, vegetation, sky, landscape, fruit, etc.), and olfactory elements (fresh air, soil, phytoncide, etc.).

The results of the χ² test conducted to see whether there was a significant association in the type of forest healing factors according to the groups are presented in Table 2. As shown in Table 2, self-guided forest therapy was significantly associated with the auditory element as forest healing factors (χ² = 8.977, *p* < 0.01). However, no other forest healing factors were associated with this type of forest therapy activities. These results show that in the self-guided forest therapy groups, auditory elements such as “bird sound” and “sound of stepping on fallen leaves” acted as healing factors for participants. These auditory elements might give comfort and pleasure to participants in a self-guided forest therapy group.

The following are examples from the essays of the participants who experienced the self-guided forest therapy.


*“We walked along the forest road. As a person who likes to walk usually, I was interested in taking a walk often. Every day, I listened to loud noises from outside, but now I walked listening to natural sounds such as birds and wind in the forest, making me feel better and smiling.”*
(SGFT participant 12)


*“The sound of birds and wind in the forest lingered in my ears. It is a sound that can be heard only in the forest at the same time. Listening to such vivid sounds made my mind feel much more relaxed, and my head felt cool.”*
(SGFT participant 15)

### 3.2. Quantitative Comparison of Health Benefits in Self-Guided Forest Therapy and Guided Forest Therapy Program

The types of health benefits derived from open coding in this study were shown in 5 categories and 44 keywords: change of mind and body, change of emotion, social interaction, cognitive change, and introspection (Table 3).

The results of the χ^2^ test conducted to see whether there was a significant association in health benefits according to the groups are presented in Table 4. As shown in Table 4, self-guided forest therapy was significantly associated with introspection (χ^2^ = 7.569, *p* < 0.01) as a health benefit. On the other hand, the guided forest therapy program was significantly associated with change of emotion (χ^2^ = 9.971, *p* < 0.01) and social interaction (χ^2^ = 21.360, *p* < 0.001). However, no other health benefits such as change of mind and body and cognitive change were associated with the type of forest therapy.

These results show that self-guided forest therapy provided an opportunity to reflect on oneself and organize ideas through dialogue with the inside. Examples of the introspection of participants in self-guided forest therapy group are as follows.


*“The best thing about running a self-guided forest therapy program was that I could organize my thoughts. I like to answer the question thrown. What about you, not the kind of question that solves the problem? A question of feeling. I could get one step closer to myself in the woods and trees, and I think it was precious. At the time, I focused solely on myself.”*
(SGFT participant 4)


*“During the self-guided forest therapy program, I was able to think deeply about what I thought, what I wanted, and what I lacked because I could lean on trees in the forest without any pressure or burden.”*
(SGFT Participant 16)

On the other hand, participants in the guided forest therapy program group experienced positive emotions due to the beauty of the forest and various elements in the forest and showed that negative emotions such as anxiety and tension were resolved. In particular, participants who experienced the guided forest therapy program mentioned many keywords related to fun/joy/laughter among positive emotions.

The following are examples of the fun/joy/laughter of participants in the guided forest therapy program group.


*“I don’t think I’ve ever laughed and focused like this lately. I laughed and enjoyed many activities with forest therapy instructors and friends, and naturally, negative feelings disappeared. Now I wait for the most time of the week for the guided forest therapy program. However, not just me but also friends who participate in the program wait for the forest therapy program the most.”*
(GFTP Participant 10)


*“It is a good time because it brings peace and stability through the forest therapy program, and it is nothing but laughter and happiness. I don’t laugh much, but I feel relaxed, so I laugh a lot these days even though it’s nothing special. It’s also because of this time.”*
(GFTP Participant 14)


*“I haven’t laughed without thinking recently, but at this time, I laughed without thinking and felt refreshed.”*
(GFTP Participant 11)

In addition, the guided forest therapy program provided participants with social effects such as building intimacy and improving interpersonal relationships. Most participants noted that forming social relationships with other participants was one of the most valuable guided forest therapy programs.

The following are examples of participants who experienced the guided forest therapy program.


*“Through hand massage activities, I felt better because the smell of oil was good, and I had a chance to touch my friend’s hand, so I learned how my friend’s hand looked and felt, and I felt more intimate with my friend.”*
(GFTP Participant 7)


*“In the first week, it was an interview activity to recognize each other in the forest. I’ve only been acquainted with my classmates, and I haven’t been so close. This activity allowed me to get closer to my partner.”*
(GFTP Participant 14)


*“I covered my eyes with an eye patch and played a game of running around the lawn with only the leader and team members. As I closed my eyes, the team members were the only ones to rely on. I was scared, but I relied on it that much, and I gained intimacy.”*
(GFTP Participant 15)

## 4. Discussion

This study provides empirical support for the difference between being in the forest alone and the guided forest therapy program. These results revealed similarities and differences between forest healing factors and health benefits that acted on the self-guided forest therapy and guided forest therapy programs. This study found that forest healing factors commonly had four categories: auditory, visual, tactile, and olfactory. In addition, this study found that the health benefits of forest therapy commonly had five categories: change of mind and body, introspection, change of emotion, cognitive change, and social interaction.

### 4.1. Self-Guided Forest Therapy

This study revealed that the self-guided forest therapy group significantly mentioned more auditory elements than the guided forest therapy program group. In particular, the self-guided forest therapy group often mentioned the sound of birds and stepping on fallen leaves. Therefore, it is thought that subjects who participated in the self-guided forest therapy group experienced healing factors through auditory elements, such as the sound of birds and stepping on fallen leaves.

The results of this study are consistent with previous findings investigating preference for natural sounds [71,72] and psychological responses [73,74]. Previous studies have reported that while artificial sounds are considered unpleasant, forest sounds like bird sounds are perceived as pleasant [75,76,77]. For example, Liu et al. [72] reported that park users prefer natural sounds to artificial sounds. Zhang [73] showed that natural sounds positively affected the recovery of individual attention, while artificial sounds inhibited the recovery of individuals. A similar study by Zhang et al. [74] reported that natural sounds such as bird sounds and water sounds positively affected the recovery of individual attention. Self-guided forest therapy is thought to have been used for its healing factors, such as birds and fallen leaves, because participants can quietly contemplate by doing forest healing on their own and listen thoroughly to the sounds of nature around them.

This study revealed that the self-guided forest therapy group significantly mentioned introspection more than the guided forest therapy program group. Nature, such as in a forest, provides an opportunity for self-reflection. Those results provide supporting evidence for the attention restoration theory. Attention restoration theory (ART) argues that it could be helpful because there are aesthetic advantages to nature, such as forests, and suggests that spending time in nature gives individuals a chance to reflect on themselves and time to think about unresolved issues [78,79,80]. Hendee and Barown [81] also reported that forest experience provides an opportunity to evaluate individual qualities as a first step toward personal growth and increases awareness of individuals.

This study indicates that being alone in nature, such as a forest, can only help immerse oneself in it. Kaplan [82] reported that being alone in the forest might give people a chance to meditate on who they were, who they wanted to be, and realize what was important to them. In a further study, Kaplan and Kaplan [78] reported that the forest’s attention recovery phase went through four stages. The first step was to organize your thoughts in your mind; the second was to regain your attention; the third was to reduce distractions in the mind, paying attention to an idea in silence. The final step was to have a chance to recover attention and have a positive change of mind through the process of setting priorities and possibilities of life, actions, and goals, including reflection on one’s life. Oh et al. [70] revealed a six-step psychological mechanism that causes emotional and cognitive changes in nature-based healing processes. It reported that participants could focus entirely on themselves and look into their emotions and thoughts when they have a deep interaction with nature. By communicating with oneself in nature, one can realize the cause of problems and determine the change, which is reported to occur mainly in tranquility when alone in nature. In addition, self-guided forest therapy is thought to create an experience of immersion by conducting activities freely on one’s own. For example, Mannel et al. [83] conducted a study of whether higher immersion levels were associated with freely selected behavior. The result showed that free-selected intrinsic motivation behavior caused positive effects such as more frequent immersion experiences, low tension, and high concentration. Therefore, the results of this study suggest that subjects who participated in the self-guided forest therapy had the opportunity to focus more on their thoughts and organize their thoughts, not on the outside.

### 4.2. Guided Forest Therapy Program

The results of this study revealed that the guided forest therapy program frequently mentioned positive emotions such as fun/joy/laughter, improvement of mood state, happiness, sense of accomplishment, and negative emotions such as reduced anxiety and reduced tension. These findings are consistent with studies suggesting that forest therapy programs improve mood states and reduce anxiety and tension [42,45,46]. In particular, there have been many mentions of positive emotions among changes in emotions, which can be a decisive factor in the relationship between happiness and positive life outcomes [84], depending on the cognitive model of coping [85]. Positive emotions can help individuals relax psychologically and restore their stress-deprived resources [86]. Positive emotions can also facilitate thinking and creativity to help solve daily life problems more flexibly [87].

In the study results, participants in the guided forest therapy program mentioned a lot of fun/joy/laughter among positive emotions. These results were judged to impact the guided forest therapy program’s characteristics and activities content. The guided forest therapy program consists of activities in which participants can experience forest therapy and interact with each other. According to Reis et al. [88], the pleasure experienced when interacting with others, especially friends, has a positive effect (in terms of fun) over fun alone. Therefore, interaction with people provides social signals of joy and pleasure, strengthening individuals’ emotional experience dynamically [89,90,91]. Moreover, it is thought that fun/joy/laughter was mentioned during guided forest therapy program activities because of play activities with natural objects such as the forest fork dance, natural five-sense game, and forest orienting.

Many of the types of benefits mentioned in the guided forest therapy program group were social interactions such as intimacy formation and improvement of interpersonal relations. The results of this study are consistent with previous findings on the social effects of forest therapy programs. For example, Cho et al. [53] reported that a two nights and three-day forest experience program improved low-income children’s sociality. Oh et al. [92] also reported that the forest therapy program enhances youth capabilities, including interpersonal skills. In addition, Jeoun et al. [93] reported that a six-session forest experience program positively affected interpersonal relations for maladjusted soldiers. Hendee and Barown [81] reported that collective forest experiences could provide social interaction and improve ties. The more individuals share their inner world and feelings with others, the more cohesive the group becomes, the more open individuals begin to be to each other [94]. The process of psychological change caused by building social relationships in the forest environment was similar to developing positive emotions through mutual understanding and consideration [60]. These results can be provided as evidence to confirm the positive impact of participating in forest therapy programs on social interactions. Therefore, this study suggests that the guided forest therapy program allows participants to feel positive emotions inside and expect social benefits through interactions between participants as they encounter various activities using the forest environment under the guidance of a forest therapist.

### 4.3. Limitations

The study had several limitations. Firstly, the participants for this study were limited to healthy university students in their 20s. To generalize the findings, further studies are needed using different groups of the population with other socio-demographic characteristics. Secondly, this study was based on self-reported essays written after forest therapy activities. Further studies are needed to compare the physiological and psychological effects of self-guided forest therapy and guided forest therapy programs using objective evaluation tools. Third, inter-rater reliability was not calculated to evaluate our coding scheme and some comments can be interpreted differently. Fourth, this study was not considered in the context of the fundamental differences between self-guided forest therapy and guided forest therapy program. For example, the guided forest therapy activities included various physical activities that would not have been possible to engage in alone. However, self-guided forest therapy activities consisted of activities that can be easily done alone. Further study is needed to analyze the effects of self-guided forest therapy and guided forest therapy by equating the contents of the program. Fifth, this study does not include a control group performing activities in a non-forest environment. Further study could also involve a better-controlled experiment using non-forest-related social activities for comparison. This study contributed to understanding differences in characteristics in self-guided forest therapy and guided forest therapy programs despite these limitations.

## 5. Conclusions

This study revealed the forest healing factors and health benefits that acted through self-reported essay analysis of participants in self-guided forest therapy and guided forest therapy programs. The results showed that the forest healing factors were divided into four categories in common: auditory, visual, tactile, and olfactory. The health benefits of forest therapy were divided into five categories: change of mind and body, introspection, change of emotion, cognitive change, and social interaction. As a characteristic of each activity, self-guided forest therapy provides an opportunity for self-reflection to focus on and think about one’s inner self. On the other hand, guided forest therapy programs have been shown to be effective in providing participants with positive emotional changes and promoting social bonds through interaction with others. These findings can provide customized forest therapy activities centered on users. Since forest healing activities can be selected according to specific groups and specific purposes, it presents the possibility of utilizing forests in various ways to promote health.

## Figures and Tables

**Table 1 ijerph-18-06957-t001:** Themes and activities of the guided forest therapy program.

Session	Themes	Activities
1	Rapport building	Ice breaking introduction; familiarity with forest; lecture on stress management
2		Clapping exercise; forest folk dance
3	Stress reduction	Forest orienteering (using natural objects to solve group mission); physical stimulation for relaxation
4		Group gaming activities using natural objects (drawing natural objects, hit the target with an acorn)
5	Improvement of sense of belonging and self-esteem	Forest exercise (forest walking, stretching)
6		Barefoot walking in forest; talking to nature
7	Cooperation and trust	Natural object five senses game; photo healing (taking pictures of nature and story-telling)
8		Forest band exercise; rope game

**Table 2 ijerph-18-06957-t002:** Written response category and frequencies on forest healing factors by type of forest therapy activity.

Category	SGFT (*n* = 57)		GFTP (*n* = 54)					
	N	%	N	%	χ²	*p*	Cramer’s V	Effect Size
Auditory element	27	47.4	11	20.4	8.977	0.003 **	0.284	Moderate
Tactile element	13	22.8	18	33.3	1.526	0.217	0.117	Weak
Visual element	12	21.1	14	25.9	0.367	0.545	0.058	Negligible
Olfactory element	11	19.3	10	18.5	0.011	0.917	0.010	Negligible

Notes: SGFT, self-guided forest therapy and GFTP, guided forest therapy program. ** *p* < 0.01.

**Table 3 ijerph-18-06957-t003:** Written response category and keywords on health benefits by type of forest therapy activity.

Category	Subcategory	Keyword
Change of mind and body	Change of mind	Relaxation of mind
		Stress relief
		Vitality/vigor
		Recovery from fatigue
		Freshness
		Self-esteem enhancement
		Competitive
	Change of body	Relaxing knotted muscles
		Clearing the head
		Physical health promotion
		Deep sleep
		Visual comfort
		Circulation of the blood
		Increase flexibility
		Exercise effect
Change of emotion	Positive emotion	Fun/joy/laughter
		Improvement of mood state
		Happiness
		Sense of accomplishment
		Positive feeling
		Appreciation
	Relieve negative emotion	Reduced anxiety
		Reduced tension
Social interaction	Relationship formation	Intimacy formation
		Improvement of interpersonal relations
		New idea of a relationship
	Understanding and mutual respect	Empathy
		Attentive attitude
		Cooperative spirit enhancement
Cognitive change	Attention restoration	Concentration enhancement
		Rest
		Healing
		Observation
		Feeling of uniting body and mind
	Memory	Innocence of childhood
		Recollection of past
Introspection	Self-awareness and reflection	Looking back on one’s life
		Self-reflection
	Explore for one’s inner self	Self-discovery
		Focus on me completely
		Have an inner conversation with oneself
		Find a way to comfort me
	Thought and mind arrangement	Thinking arrangement
		Disappearance of miscellaneous thought

**Table 4 ijerph-18-06957-t004:** Written response category and frequencies on health benefits by type of forest therapy activity.

Category	SGFT (*n* = 57)		GFTP (*n* = 54)					
	N	%	N	%	χ²	*p*	Cramer’s V	Effect Size
Change of mind and body	44	77.2	46	85.2	1.155	0.283	0.102	Weak
Introspection	36	63.2	20	37.0	7.569	0.006 **	0.261	Moderate
Change of emotion	22	38.6	37	68.5	9.971	0.002 **	0.300	Moderate
Cognitive change	21	36.8	22	40.7	0.178	0.673	0.040	Negligible
Social interaction	5	8.8	26	48.1	21.360	<0.001 ***	0.439	Relatively strong

Notes: SGFT (‘Self-guided’ forest therapy) and GFTP (guided forest therapy program). *** *p* < 0.001, ** *p* < 0.01.

## Data Availability

The data presented in this study are available on request from the corresponding author. The data are not publicly available due to privacy.

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
