# Peer review of "Forest Therapy Alone or with a Guide: Is There a Difference between Self-Guided Forest Therapy and Guided Forest Therapy Programs?"

_ijerph, 2021, doi:10.3390/ijerph18136957_

Round 1

Reviewer 1 Report

The study has great value. It appears to be conducted well and on an interesting, important, and understudied topic.

However, I'm simply confused on what data were collected and how they were analyzed. It looks like qualitative data were collected and coded but then somehow quantitized based on themes around forest healing factors. Is that correct?

The paper's methods and results need substantial reorganization and greater description in my opinion. At this point, I cannot conduct a robust review of the Results section without knowing where these data came from.

Minor comments: 

  • Inconsistent tense in methods
  • "Different forest healing factors" is jargon - could it be replaced in the objectives? Further, should walking along vs guided be explicit in the study objectives?
  • Grammar and english writing is awkward in some places throughout the manuscript
  • "Self-forest therapy" is awkward - how about "self-guided forest therapy"?
  • When were guided therapy sessions?
  • "assay" misspelled
  • Were 38 students randomly assigned to one condition or the other? Could "Between" or "within" subjects design be explicit? A study timeline figure would be helpful. I'm unclear when the 8 weeks took place, and if there are more than 8 weeks
  • Were qualitative data coded by a single researcher? Was coding emergent?
  • Quantitative data analysis section in methods is unclear since no surveys/survey batteries were being described as administered earlier.Again, I don't know what "forest healing factors" are.

Author Response

Reviewer 1#

Thank you so much for the detailed comments. We modified and added content on research methods and data collection methods as a whole.

  1. We added a clear research objective (Line 145-156) and added research methods (Line 213-227; 235-239; 251-254).
  2. We modified grammar and English writing throughout the manuscript.
  3. We modified ‘self-forest therapy’ to ‘self-guided forest therapy’.
  4. We added information on when the guided forest therapy program started (Line 184-185).
  5. We modified “assay” to “Essay”.
  6. We added that we did randomly assigned into two groups (Line 165).
  7. Our study was conducted once a week for eight weeks, totaling eight sessions. The forest activities interventions were provided from September to November of 2019.
  8. We carried out four stages of open coding with other authors (Line 237-239).
  9. We added a description of the quantitative analysis (Line 251-254). Forest healing factors classify forest environmental factors that participants feel during interventions. Examples include sunlight, sound, soil, air, and phytoncide, etc. During the analysis of the essay, these forest environmental factors were remarkably expressed and mentioned. So, we compared the differences between the two groups with the frequency of forest healing factors derived from the essay.

Reviewer 2 Report

Thank you for the opportunity to review your paper. My comments are attached. I'm only able to upload one file so I'm not able to upload the marked up manuscript here as I have indicated in my comments. I will send to the editor with a request to pass along to you. 

kind regards. 

Author Response

Reviewer 2#

Thank you so much for the detailed comments. We modified and added content on research methods and data collection methods as a whole.

  1. We modified the ‘to derive’ to ‘to investigate’ (line 10).

  1. We modified the context (i.e., Thirty-seven undergraduate students participated in a randomized experiment) (line 12-13).

  1. We revised the number of participants. At first, our study had a total of 38 participants. However, during the study, one participant dropped out of the guided forest therapy program group. So, the sample size of this study is 37 undergraduate students. Each participant was asked to submit three essays during the eight sessions. Essays were collected three times after completing one to three sessions, four to six sessions, and seven to eight sessions, respectively. As a result, 111 essays were analyzed, with 57 essays submitted by 19 self-guided forest therapy groups and 54 essays submitted by 18 guided forest therapy groups (See Line 213-222).

  1. We added the research method to performed qualitative analysis (Line 224-227; 235-239).

  1. We revised following sentence the for clarity: “guided forest therapy programs provide positive emotional changes and promoting social bonds through interaction with others.” (Line 24-25)

  1. We add Lim et al. (2020)'s study as an example of a direct comparison between guided and unguided forest therapy. (Line 139-141)

  1. We added the following explanation: All 37 participants (19 self-guided forest therapy participants and 18 guided forest therapy program participants) engaged in each of the 8 sessions together. (Line 187-188; 200-201)

  1. We added the length of the self-guided forest therapy trail and route form. (Line 204). Also, we added the overall area of the site location (Line 175).

  1. We added about the reason for selecting qualitative analysis (Line 224-227)

  1. We modified non-directional hypotheses to directional hypotheses (Line 148-156).

  1. We modified “Self-forest therapy activities were designed and distributed as the same themes of the forest therapy program described earlier” to “Self-guided forest therapy activities were designed and distributed for the same purpose as the forest therapy program described earlier” (Line 205-208). It means having the same purpose as a guided forest therapy program. In other words, the purpose of the self-guided forest therapy is also to reduce stress and improve self-esteem and happiness for participants.

  1. We added a description of the data collection method (Line 213-222).

  1. Thank you so much for the detailed comments. The reason we did quantitative analysis is to conduct a quantitative comparison between groups according to the frequency of words related to forest healing factors and health benefits. The data source for this analysis is a quantified value of forest healing factors and health benefits derived through qualitative analysis of essays submitted after the interventions.

  1. We deleted previous Table 2 (Written response category and keywords on forest healing factor by type of forest therapy activity) because there were many duplicates in Table 2 and Table 3 and the text.

  1. Thank you so much for your kind comment. We modified 3.1 (Miss-number). However, we didn’t modify the p-value of 0.000 should be written as <.001 in Table 5. This is because the text indicates a range of statistical significance levels (i.e., p<0.001), and Table 5 shows the actual value.

  1. We acknowledge that there is a fundamental difference between the two conditions. So, we added these points to the limitation in section 4.3 (Line 464-470).

  1. We added a conclusion on the potential for the findings of this study (Line 485-488).

  1. We added the lack of control groups in section 4.3 (Line 470-472).

Round 2

Reviewer 1 Report

I reviewed this paper before. The authors have improved it and I can now understand what they did. However, I still think the paper would be much stronger and clearer if the hypotheses were better founded and the results were aligned with these hypothesis testing. Specific comments below

New RQ are an improvement for the paper. However, they are all grammatically incorrect.

Methods section contain some present tense and some past tense verbs. Recommend consistency.

The revisions suggest the 'self-guided' therapy sessions were performed in a group, as were the guided sessions. The 'self-guided' labeling makes it sound to this reader that they were individual. Recommend alternative language with word 'group.'

New text "Because forest therapy proceeds through a very complex mechanism, existing quantitative research methodologies alone are limited in analyzing multiple aspects of the forest therapy process. Therefore, qualitative studies are needed to complement quan-titative studies." is very arguable. There are dozens of forest bathing studies with quantitative measures. Further, the first phrase in this new text is unclear. I'm not arguing at all that qualitative methods are not appropriate here, I just don't think this is an accurate or strong rationale as is.

I belive what you are doing is 'quantitizing' the qualitative data.

Since inherently this was a qualitative study - since the data the authors collected was indeed qualitative - then it is most typical and appropriate to lead the Results section with a description of the qualitative coding results. Not jumping into the counts/frequencies of codes.  In particular, a thorough description of each code/theme is needed with example quotes.

Strongly recommend against the acronyms used in the tables. These are annoying to have to look up.

The Results sections do not align with stated hypotheses. Further concerning is that the hypotheses are not clear or sufficiently discussed in the introductio (i.e., what is "reflect on onseself" mean? and what is the lit review/evidence for hypotheszing this?)

Author Response

Reviewer 1#

Thank you so much for the detailed comments. We modified and added content on research methods and data collection methods as a whole.

  1. We modified grammar in New RQ and matched the current tense with the past tense. (Line 173-180).
  2. We replaced the ‘self-guided label with the word ‘group’.
  3. We deleted the new text “Because forest therapy proceeds through a very complex mechanism, existing quantitative research methodologies alone are limited in analyzing multiple aspects of the forest therapy process. Therefore, qualitative studies are needed to complement quantitative studies.” And we added to the content of qualitative data analysis (Line 260-289).
  4. We added the description of each code/theme (Line 275-289).
  5. Thank you so much for the detailed comments. There are some parts where it is difficult to express the contents of the table with abbreviations. However, we adjusted the spacing of the table so that it can be seen at once (Line 336).
  6. We added the introduction to fully discuss the stated hypothesis (Line 88-89/95-108/104-153).

Reviewer 2 Report

Thank you for the opportunity to review your revised paper.
Written expression remains problematic throughout.
In addition to matters of written expression, I have some remaining concerns about the study as follows:
2.3 Experimental design. It appears that were some further inherent differences between the two interventions beyond simply guided or self-guided. For instance, there is a 30-minute difference in duration, and the guided group engaged in more social and collective activities requiring active engagement (e.g., clapping, folk dancing, group gaming, forest band, rope game, etc.). On the other hand, the self-guided group engaged in what might be considered relatively more passive and less social activities (e.g., respiration, walking, meditation, and exercise). These differences don’t lend themselves to a fully controlled experiment (I note that you’ve acknowledged this as a limitation in section 4.3, although not in as detailed a fashion as I think the differences warrant).
2.4.2 Qualitative data analysis. While some detail has been provided, there remains insufficient information about the choice of qualitative data analysis approach. For instance, which paradigms or philosophical positions were followed? What were the ontological, epistemological, and theoretical perspectives of your methodology? Further, what steps (if any) were taken to avoid or reduce researcher bias? The authors performed the coding; was there any attempt to blind the authors to which groups the essays came from?
Table 4. I reiterate my point about reporting of p values, which should not be reported as 0.000. If the value is too small to accommodate the 3 decimal points, the value should still be written as <.001.
Quantitative data analysis. I reiterate my point about chi-square analysis. The chi-square test of independence is test of association, not difference, and the correct way to report non-significant outcomes is to say that X was independent of Y, or that no association was found between X and Y. For all reports of chi-square tests, effect sizes should be included. 

Author Response

Reviewer 2#

Thank you so much for the detailed comments. We modified and added content on research methods and data collection methods as a whole.

  1. Thank you so much for the detailed comments. We acknowledge that there is a fundamental difference between the two conditions. We added these points to the limitation in section 4.3. In subsequent studies, we will conduct fully controlled experiments by minimizing the inherent differences in these interventions.
  2. We added to the qualitative data analysis approach of this study and added to the content measures to prevent researcher bias (Line 260-268).
  3. We revised the representation of the results of the chi-square test (Line 311-315/ 338-344).
  4. We modified the p-value expression and added the effect size (Table 2 & Table 4).
